# Unveiling Major Depressive Disorder Through TMS-EEG: From Traditional to Emerging Approaches

**DOI:** 10.3390/biomedicines13102474

**Published:** 2025-10-11

**Authors:** Antonietta Stango, Claudia Fracassi, Andrea Cesareni, Barbara Borroni, Agnese Zazio

**Affiliations:** 1Neurophysiology Lab, IRCCS Istituto Centro San Giovanni di Dio Fatebenefratelli, 25125 Brescia, Italy; 2Psychiatry Unit, IRCCS Istituto Centro San Giovanni di Dio Fatebenefratelli, 25125 Brescia, Italy; 3Molecular Markers Laboratory, IRCCS Istituto Centro San Giovanni di Dio Fatebenefratelli, 25125 Brescia, Italy; 4Department of Clinical and Experimental Sciences, University of Brescia, 25121 Brescia, Italy

**Keywords:** transcranial magnetic stimulation (TMS), electroencephalography (EEG), depression, TMS-evoked potentials (TEPs), time-frequency analysis, source analysis, machine learning, microstates

## Abstract

Major depressive disorder (MDD) is one of the most prevalent psychiatric conditions and is characterized by alterations in cortical excitability, network connectivity, and neuroplasticity. Despite significant progress in neuroimaging and neurophysiology, the identification of objective and reliable biomarkers remains a major challenge, limiting diagnostic accuracy and treatment optimization. Transcranial magnetic stimulation combined with electroencephalography (TMS-EEG) has emerged as a powerful methodology to probe causal brain dynamics with high temporal resolution. This review aims to summarize recent advances in the application of TMS-EEG to MDD, highlighting the transition from traditional TMS-evoked potential (TEP) analyses to more advanced, multidimensional approaches. We reviewed original research articles published between 2020 and 2025 that investigated neurophysiological markers and approaches to MDD using TMS-EEG. Traditional TEP measures provide markers of local cortical responses but are limited in capturing distributed network dysfunction. Emerging approaches expand the scope of TMS-EEG, allowing for the characterization of oscillatory activity, connectivity patterns, and large-scale network dynamics. Recent contributions also demonstrate the potential of computational and multivariate techniques to enhance biomarker sensitivity and predictive value. Taken together, recent evidence highlights TMS-EEG as a uniquely positioned methodology to investigate the neurophysiological substrates of MDD. By linking conventional TEP-based indices with innovative analytic strategies, TMS-EEG enables a multidimensional assessment of cortical function and dysfunction that transcends traditional descriptive markers. This integrative perspective not only refines mechanistic models of MDD but also opens new avenues for biomarker discovery, patient stratification, and treatment monitoring. Ultimately, the convergence of advanced TMS-EEG approaches with clinical applications holds promise for translating neurophysiological insights into precision psychiatry interventions aimed at improving outcomes in MDD.

## 1. Introduction

Depression has been the primary contributor to the global burden of mental disorders over the past three decades. Major depressive disorder (MDD), the most severe form of depression, is classified as an episodic mood disorder marked by the occurrence of one or more major depressive episodes. Findings from a recent cross-national investigation indicate that the average lifetime prevalence of MDD is 7.5% among men and 13.6% among women [1].

Recent epidemiological estimates indicate that approximately 5.7% of the adult world population currently suffers from major depression [2]. A depressive episode is defined as a period of at least two weeks characterized by persistent low mood and/or anhedonia (loss of interest or pleasure), accompanied by additional symptoms such as changes in sleep or appetite, fatigue or loss of energy, impaired concentration, feelings of worthlessness or excessive guilt, psychomotor agitation or retardation, and recurrent thoughts of death or suicidal ideation. The disorder significantly interferes with psychosocial functioning, impairing interpersonal relationships, occupational performance, and academic achievement [3]. A recent study offers updated estimates of the economic burden associated with MDD, accounting for both direct and indirect costs. In the United States alone, the societal cost was projected at $382.4 billion in 2023, corresponding to an average of $16,854 per adult with MDD [4].

While epidemiological and clinical data highlight the immense personal and societal impact of MDD, these observations only partially capture the complexity of the disorder. Advances in neurobiological research have revealed that MDD is not limited to disturbances in mood and behavior but is also associated with profound alterations in brain structure and function. Understanding these neurobiological underpinnings is essential, as they not only provide insight into the mechanisms driving symptom expression but also inform the development of novel diagnostic tools and therapeutic strategies.

Neurobiological studies have shown that MDD is associated with widespread functional, structural and pathophysiological abnormalities in the brain, chronic episodes resulting in progressive neuroanatomical changes [5,6]. These alterations include cortical changes secondary to hypoperfusion, neuroinflammation, and disruptions in network-level connectivity, particularly within the dorsolateral prefrontal cortex (DLPFC). Beyond the DLPFC, converging evidence points to abnormalities in the sensory-motor network, which have been linked to psychomotor alterations frequently observed in MDD [7]. Limbic regions such as the hippocampus and amygdala also play a critical role: stress-related hypercortisolemia has been shown to impair neurogenesis, contribute to hippocampal dysfunction, and induce structural and functional alterations in fronto-limbic circuits [8]. Although a consistent amount of evidence has shown the profound neurological effects of MDD on the human brain, and despite several mechanistic pathways have been proposed, the etiology of MDD remains incompletely understood [6]. Moreover, MDD diagnosis relies primarily on clinical interviews and standardized symptom questionnaires [9]. However, the identification of reliable biomarkers, defined as objective, quantifiable indicators of biological or pathological processes, remains a critical step toward improving diagnostic accuracy, prognostic evaluation, and treatment monitoring [10,11].

In this context, the combined use of transcranial magnetic stimulation and electroencephalography (TMS-EEG) emerged as a promising approach for biomarker discovery in MDD [5,6,7]. While TMS allows for direct perturbation of a cortical area bypassing sensory processing, therefore allowing for causal inferences, EEG records the spread of cortical activation with excellent temporal resolution via scalp electrodes [12,13]. By integrating perturbation and recording, TMS-EEG offers a unique window onto the dynamic interactions underlying MDD pathophysiology, providing clinically relevant information that complements traditional assessments [14].

The left DLPFC is the most frequently targeted region in MDD, given its central role in the disorder’s pathophysiology [15]. Accordingly, the DLPFC is routinely selected both for TMS-EEG recordings and for repetitive TMS (rTMS) protocols aimed at inducing long-lasting plasticity changes [16].

This review article provides an overview of the most recently published works involving TMS-EEG methodology in the study of MDD from 2020 to 2025 (last search performed on 30 August 2025). It first outlines conventional measures based on TMS-evoked potential (TEP) components and then focuses on more advanced metrics and approaches that exploit the multidimensional information provided by TMS-EEG. Rather than providing an exhaustive systematic synthesis, this review aims to capture recent conceptual trends and methodological developments in the field. As an overview, Table 1 and Table 2 summarize the included studies and their essential details. Table 1 focuses on traditional TMS-EEG metrics and Table 2 on emerging TMS-EEG approaches. Both tables report information on sample composition (patients with MDD or treatment-resistant depression—TRD—and healthy controls—HCs), stimulation sites, medication status, and coil type and stimulation intensity (expressed in % of the resting motor threshold—rMT). These tables serve multiple purposes: they allow readers to quickly compare studies, highlight methodological variability, and illustrate potential sources of heterogeneity that may influence the interpretation of results. Figure 1 shows the amount of papers identified for year and their distribution between traditional and advanced approaches.

## 2. Traditional TMS-EEG Measures in MDD

Early applications of TMS-EEG methodology in MDD have primarily relied on conventional indices, such as TEPs. TEPs are obtained by averaging multiple TMS pulses in the time domain and are characterized by a sequence of positive and negative deflections typically analyzed from ~20 ms to several hundred milliseconds after stimulation [34]. TEPs are thought to reflect the spread of cortical activation from the stimulated area to interconnected regions, thereby providing a non-invasive proxy of effective connectivity and network excitability [12,13]. The most commonly investigated features of TEPs are the amplitude and latency of specific peaks, usually extracted from predefined regions of interest around the stimulation site at the sensor level. Other traditional measures include global or local mean field power/amplitude (GMFP/A and LMFP/A, respectively), which summarize the overall strength of the evoked response across channels or within a region of interest, often complemented by measures such as the GMFA area under the curve [35].

Alterations in TEP components have been revealed in MDD and other psychiatric disorders, suggesting their utility as biomarkers of disease-related changes in cortical excitability and inhibition [14,36]. A comprehensive synthesis of TEPs findings in MDD, including their potential as treatment-response biomarkers, falls beyond the scope of the present work and can be found in recent reviews [9,37]. In the present section, we briefly recall some of the main findings from this literature and highlight more recent contributions that extend or refine previous evidence (Table 1).

Specifically, following DLPFC stimulation, greater amplitude of N45 and N100 components, commonly associated with inhibitory processes, have been observed in MDD compared to controls, alongside elevated cortical reactivity, as indexed by the GMFA area under the curve [17,35]. In addition, other TEPs components such as P60, N180, and P/N280 have been preliminarily explored as potential diagnostic biomarkers, showing alterations compared to controls or associations with depressive symptoms, although their underlying neurophysiological mechanisms remain less well understood [22,23,25,35].

Finally, the treatment-related modulation of TEP components has been investigated, highlighting the potential of TEPs as predictive biomarkers capable of capturing cortical changes induced not only by rTMS [19,24] and theta-burst stimulation [16,18,21], but also by other non-TMS interventions [20].

The use of TEPs-derived measures as diagnostic and predictive biomarkers for MDD presents some advantages. They rely on a steadily growing body of literature that has contributed to clarifying the physiological mechanisms underlying different TEP components (e.g., refs. [38,39]), facilitating interpretation and supporting cross-study comparisons. Moreover, the relative simplicity of extracting amplitude and latency from well-characterized peaks makes these indices relatively accessible.

However, conventional TEPs-derived measures face important limitations. By concentrating mainly on electrodes near the stimulation site and on a few predefined peaks, they capture only a fraction of the multidimensional information contained in the TMS-EEG signal. Consequently, conventional analyses provide only a partial view of cortical dynamics, leaving aspects such as the broader spatial distribution of activity, trial-to-trial variability, and more complex signal patterns underexplored. As a result, while amplitude and latency of canonical TEP components provide a convenient and interpretable readout, they offer a limited window on the complexity of the neural processes engaged by TMS. These constraints have motivated the development of more advanced analytical approaches that aim to exploit the full spatiotemporal and spectral richness of the TMS-EEG signal, with the goal of yielding more comprehensive and clinically meaningful insights into MDD pathophysiology.

## 3. Emerging TMS-EEG Metric and Approaches in MDD

Building on foundational insights from traditional TMS-EEG measures such as TEPs, recent methodological advances have expanded the investigation of cortical dynamics in MDD [23,37]. Unlike conventional indices, which primarily examine peak amplitude and latency at electrodes near the stimulation site, or give a simplified measure of global cortical reactivity, emerging approaches exploit the spatiotemporal and spectral information provided by TMS-EEG. These innovations have facilitated the development of novel metrics and putative biomarkers, contributing to a more precise characterization of MDD [9]. They hold promising translational potential, including personalized treatment strategies, monitoring therapeutic efficacy, and supporting biologically grounded diagnostic approaches [9].

Emerging analytical frameworks allow for the assessment of cortical excitability and connectivity across multiple dimensions, including frequency-specific oscillatory activity, large-scale network dynamics, trial-by-trial variability, and source-localized cortical generators. The integration of TMS-EEG with multimodal neuroimaging, combined with machine learning approaches, further enhances predictive capabilities, enabling patient stratification and individualized modeling of treatment response. Collectively, these approaches leverage the inherent advantages of TMS-EEG, its non-invasive nature and capacity to directly perturb cortical circuits, to integrate neurophysiological insights with clinical application in MDD [9].

### 3.1. Oscillatory Dynamics

Among the emerging TMS-EEG approaches, analyses of oscillatory dynamics represent one of the earliest developed techniques [34,40]. While relatively established compared to more recent metrics, these analyses exploit the multidimensional information provided by TMS-EEG, providing essential insights into frequency-specific cortical activity and large-scale network function [41].

Oscillatory power across frequency bands reflects synchronized neuronal population activity and provides a sensitive index of excitation–inhibition balance [35]. Alterations in these dynamics are closely linked to large-scale network dysfunction, processes strongly implicated in the pathophysiology of MDD [42].

While most prior work has emphasized TEPs [17,35] (see also Section 2), time–frequency analyses allow for the quantification of both phase-locked and induced responses, providing a broader characterization of cortical dynamics [43]. In this context, Hill [27] used TMS-EEG to examine oscillatory power in patients with MDD compared to healthy controls, and to evaluate modulation of these responses following convulsive therapies. Before treatments, MDD subjects exhibited increased delta, theta, and alpha power compared to healthy controls following stimulation of the dorsolateral prefrontal cortex, but no differences were observed with motor cortex stimulation, underscoring the prefrontal specificity of oscillatory abnormalities. Following treatment, magnetic seizure therapy (MST) selectively reduced delta and theta power at the DLPFC, whereas electroconvulsive therapy (ECT) induced broader reductions across delta, theta, and alpha bands at both DLPFC and M1. Importantly, a reduction in prefrontal alpha power after MST correlated with improvement in depressive symptoms. Together, these findings extend earlier evidence of aberrant cortical reactivity in MDD [17,35] by showing frequency-specific oscillatory alterations localized to the DLPFC, and they provide the first demonstration that convulsive therapies modulate TMS-evoked oscillatory power in a site- and treatment-dependent manner. This work highlights the potential of TMS-EEG oscillatory measures as candidate biomarkers for disease characterization and therapeutic monitoring in depression.

The key advantages of oscillatory analyses lie in their millisecond-level temporal resolution and capacity to distinguish between phase-locked and induced responses, offering a richer characterization of cortical function and tapping into oscillatory dynamics that are regarded as core mechanisms of neural communication.

### 3.2. Microstate Analysis

EEG microstates can be defined as unique topographies that remain stable for brief intervals of a few tens of milliseconds before shifting to another configuration, and are thought to reflect dynamic transitions between large-scale neural networks [44]. Originally developed for the analysis of resting-state EEG [45], microstate analysis has since been extended to event-related potentials [46] and, more recently, to TEPs [47,48].

Alterations in EEG microstate properties have been consistently reported across various neuropsychiatric disorders, including MDD [49,50,51]. However, to date, only one study has applied microstate analysis to TMS-EEG recordings in MDD [32].

In this study, Zhang and colleagues applied single-pulse TMS over the primary motor cortex and analyzed the resulting EEG responses using microstate analysis. They identified four microstate classes (MS1–MS4), notably showing patterns consistent with previous EEG findings, with several parameters significantly altered in MDD patients compared to healthy controls. Specifically, MDD patients showed an increased coverage of MS1 and prolonged durations of MS4, whereas MS3 exhibited reduced coverage, duration, and occurrence. The patterns of transitions among MS1, MS3, and MS4 also differed between groups, indicating a reorganization of temporal dynamics in cortical networks. Furthermore, spatial correlations between microstates were also altered, reflecting a disruption in network connectivity in MDD. Importantly, lower MS3 parameters were associated with higher scores on the Hamilton Depression Rating Scale (HAMD-24) scale and related symptom factors, highlighting its potential diagnostic and prognostic relevance. Future applications of microstate analysis on TMS-EEG data could target the DLPFC, a region more directly implicated in MDD pathophysiology.

In summary, a key advantage of the microstate framework is that it does not rely on the selection of specific electrodes but instead captures the global spatial configuration of scalp potentials while preserving the millisecond temporal resolution of EEG. When applied to TMS-EEG, microstate analysis provides a data-driven means to characterize the spatiotemporal dynamics of effective connectivity, and its combination with other neuroimaging modalities may further clarify the functional significance of different microstate patterns in MDD.

### 3.3. Trial-by-Trial Variability

Trial-by-trial neural variability (TTV) refers to the differences in neuronal responses across repeated presentations of the same stimulus. Such variability is a common feature of neural activity with important implications for understanding neural coding, behavior, and cognitive processes. Importantly, neural variability has increasingly been recognized as a core property of brain function, reflecting dynamic coding mechanisms, network adaptability, and the capacity of the brain to flexibly respond to environmental demands [52]. This perspective is particularly relevant in the context of non-invasive brain stimulation, where accounting for individual variability can enhance the precision and efficacy of stimulation protocols [53].

Conventional averaging approaches in electrophysiology emphasize the mean response across trials, but in doing so they often obscure fluctuations that may carry critical functional significance. In contrast, quantifying TTV captures the dispersion and temporal dynamics of neural activity across repetitions, offering a richer and potentially more sensitive characterization of brain function and its variability [54]. Incorporating measures of variability into event-related analyses therefore provides significant advantages, particularly for understanding individual differences and for identifying candidate biomarkers in clinical populations.

A recent paper from Niu [29] represents the first study to investigate TTV in TMS-EEG data from healthy controls and MDD patients. The authors examined whether altered neural variability constitutes a neurophysiological feature of MDD. The authors applied single-pulse TMS (110% of the resting motor threshold) over the left DLPFC while recording EEG responses in both healthy controls and patients with MDD. TTV of TEPs and oscillatory activity was quantified to assess differences in cortical excitability and network stability between groups.

Beside the analysis of conventional TEPs, the authors proposed a novel approach to quantify TTV, based on the maximum eigenvalue of the real binary correlation matrix across trials. This measure was compared against the traditional standard deviation–based method. The eigenvalue-based approach proved more sensitive and reliable in capturing variability across trials.

The results revealed abnormal TTV patterns in MDD patients, specifically, reduced variability in the gamma band (32–64 Hz) over central-parietal and right temporal regions, and increased variability in the delta band (1–2 Hz) in parietal areas. Moreover, they found that gamma-band TTV negatively correlated with clinical severity as assessed by the HAMD-17.

Together, these findings support the view that MDD is associated with a loss of neural flexibility and reduced dynamic range of cortical responses. By demonstrating that TTV derived from TMS-EEG can differentiate between patients with MDD and healthy controls, Niu et al. provide evidence that variability-based measures may represent promising biomarkers for characterizing disease-related dysfunction and potentially predicting treatment outcomes. However, the study also presents some limitations, underlining the necessity of larger, more controlled investigations to validate the robustness and clinical utility of these variability-based biomarkers.

### 3.4. Source-Level Analysis

Source-based analyses represent an important step forward in the exploitation of TMS-EEG data, as they allow for the estimation of neural activity directly at the cortical level rather than at the scalp. This approach overcomes some of the limitations of sensor-level measures, such as the influence of volume conduction and reference choice, and provides a more physiologically grounded description of TMS-evoked activity [55,56].

Several source reconstruction methods and source-derived metrics have been developed, among which significant current density (SCD) and significant current scattering (SCS) have been applied in MDD [26,57,58]. SCD quantifies the strength of cortical responses in anatomically defined regions, whereas SCS captures the spatial extent of TMS-evoked activity across cortical regions [59]. Using these measures, Hadas and colleagues have previously reported baseline hyperactivity of the subgenual cingulate cortex in MDD, which was highly accurate in discriminating patients from controls according to SCD analysis [57]. Notably, this hyperactivity was attenuated after an rTMS protocol, approaching control levels. In addition, connectivity between the DLPFC and subgenual cingulate cortex indexed by SCS was found to predict rTMS treatment response [57] and to be modulated by MST [26]. Another recent study applied the SCS metric in MDD, showing stronger connectivity between the DLPFC and other key regions, including the medial prefrontal cortex and anterior cingulate cortex, in patients with suicidal ideation compared to both healthy controls and MDD patients without suicidal ideation [33]. Together, these findings suggest that SCD and SCS capture distinct aspects of MDD pathophysiology and hold promise as diagnostic and predictive biomarkers.

Importantly, source reconstruction does not only enable novel indices but also extends existing metrics, such as TTV and oscillatory dynamics, thereby improving anatomical interpretability and reducing confounds from scalp-level projections. This flexibility positions source-based analysis as a unifying framework that links TMS-EEG signal features with neurobiological models of MDD.

### 3.5. Multimodal/Multiscale TMS-EEG Approaches

In neuroscience, a multimodal approach combines different investigation techniques, while a multiscale approach integrates multiple levels of analysis. Methodological advancements have enabled multiple simultaneous combinations of approaches, including different neuroimaging modalities (e.g., EEG and functional magnetic resonance, fMRI [60]), and the integration of stimulation and recording methods (e.g., transcranial electrical stimulation and magnetoencephalography [61]). In this context, TMS-EEG can itself be considered a multimodal approach, as it combines a brain stimulation technique with simultaneous neurophysiological recording [62], but can be further combined with additional methods. Among these, the most widely used approach involves integrating TMS-EEG with structural MRI to improve source-level estimation of neural activity (Section 3.4).

Insightful examples of multimodal and multiscale TMS-EEG applications in MDD come from the work of Wada and colleagues, which links neurophysiological, source-level, and cellular-level analyses using a virtual histology framework [28,30]. The virtual histology approach combines neuroimaging findings with post-mortem regional gene expression profiles from brain atlases to infer the contribution of specific cell types to macroscopic brain alterations. When applied alongside TMS-EEG, virtual histology provides complementary insights into how cellular architecture shapes large-scale network dynamics, offering a powerful framework to investigate MDD mechanisms across multiple levels.

Wada [28] combined TMS-EEG, MRI and virtual histology to investigate the pathophysiology of treatment-resistant depression. Interestingly, they also performed source-based time-frequency analysis, representing a clear example of combining multiple approaches and innovative metrics. They observed reduced signal propagation from the left DLPFC to the salience network in the theta and alpha bands, which was significantly correlated with oligodendrocyte-specific gene expression. In a more recent study [30], the same group employed an intracortical facilitation (ICF) paradigm, in which the outcome of a double-pulse TMS (with a specific inter-stimulus interval of 10 ms) is compared to single-pulse TMS. Typically, the ICF paradigm is performed over the primary motor cortex, and the facilitation of peripheral motor-evoked potentials measured via electromyography is thought to reflect NMDA receptor-mediated neural activity. Here, the authors applied an ICF paradigm in a TMS-EEG recording stimulating the DLPFC, and combined this approach with the MRI recordings and virtual histology. They observed a decreased source-level glutamatergic neural activity at the stimulation site in patients with treatment-resistant depression compared to controls. Additionally, they reported associations between impaired NMDA receptor-mediated propagation and astrocyte-related gene expression, highlighting a potential link between cellular-level alterations and network-level dynamics.

Multimodal and multiscale approaches offer the unique advantage of combining complementary sources of information, allowing researchers to link the high temporal resolution of EEG with the spatial precision of neuroimaging, the mechanism-based insights from cellular- or receptor-level analyses, and the causal perturbation provided by brain stimulation. Such integrative frameworks enhance our ability to characterize complex neural dynamics and their pathophysiological relevance, providing a more comprehensive and informed understanding of brain function which may be critical in the study of MDD pathophysiology.

### 3.6. Machine Learning Approaches to TMS-EEG

Machine learning (ML) methods have gained prominence in depression research, offering data-driven tools to detect subtle neurophysiological patterns, integrate heterogeneous features, and generate predictive models for diagnosis and treatment response. Several studies have demonstrated the feasibility of EEG-based ML classifiers in depression, employing a wide range of algorithms, from support vector machines and random forests to deep learning methods. Reported accuracies for predicting treatment outcomes from EEG data range between 83.1% to 97.3% in determining treatment outcomes for major depressive disorders [63,64,65].

The application of machine learning to TMS–EEG has gained increasing attention in recent years as a strategy to identify reliable biomarkers of MDD. A particularly innovative contribution was made by Noda [66], who developed an AI-based decision-support system integrating resting-state EEG and single-pulse TMS-evoked EEG. Their study included 60 patients with MDD and 60 healthy controls, with TMS applied to the left DLPFC. From the recordings, the authors extracted features spanning spectral power, phase synchronization, and phase–amplitude coupling in frontal electrodes. Nine ML algorithms were tested, including logistic regression, linear discriminant analysis, SVM, k-nearest neighbors, naïve bayes, decision tree, random forests, extra trees and lightGBM. The best performance was achieved by linear discriminant analysis applied to a combined multimodal feature set (resting-state, pre-stimulus TMS–EEG, post-stimulus TMS–EEG, and their differences), yielding a mean area under the curves of 0.922.

Thus, the Noda [31] study provides a demonstration of how ML-driven TMS–EEG analysis can advance objective diagnosis in MDD, integrating clinical decision support with insight into cortical dysconnectivity in depression.

Looking ahead, ML is expected to play a central role in personalized psychiatry, enabling individualized predictions of treatment response to TMS and pharmacological interventions. Key challenges include the need for larger and more balanced datasets, external validation, and model interpretability to ensure clinical applicability. Integration with multimodal data (e.g., MRI, cognitive and clinical scales) and the development of hybrid models combining biophysical priors with ML-based feature discovery may further enhance predictive accuracy.

## 4. Conclusions

TMS-EEG has emerged as a powerful tool in the study of brain dynamics, uniquely combining causal perturbation with the ability to track the propagation of activity across cortical networks. In recent years, several innovative approaches have been introduced in MDD research, allowing for the exploitation of the multidimensional information embedded in TMS-EEG signals. These advances include time–frequency and source-based analyses, microstate dynamics, and trial-by-trial variability measures, all of which go beyond conventional TEPs and exploit the multidimensional information from TMS-EEG. Furthermore, the integration of TMS-EEG with complementary techniques, such as structural neuroimaging and virtual histology, as well as the application of machine learning, offers further opportunities to deepen our understanding of MDD pathophysiology and potentially improve diagnostic and predictive accuracy.

However, the adoption of TMS-EEG metrics poses critical challenges that can limit their immediate translation into clinical practice. A first general limitation in the current TMS-EEG literature on MDD arise from the heterogeneity of depression, the presence of comorbidities, the confounding influence of medication and other concurrent treatments, and demographic variability, which constrain the generalizability of study findings [67]. In addition, most existing studies rely on cross-sectional or short pre–post designs, which limit causal inference and the evaluation of temporal stability. Longitudinal investigations are needed to establish whether TMS–EEG markers reflect enduring traits or dynamic, treatment-sensitive changes.

A further obstacle to clinical translation lies in the absence of robust normative databases and reliable automated analysis pipelines, a limitation that affects not only emerging but also traditional TMS–EEG metrics [68,69]. Although conventional TEP measures are often regarded as more established, the lack of standardized acquisition parameters and preprocessing pipelines still hampers their reliability and cross-study comparability. Future work should therefore focus on developing best practices to harmonize acquisition protocols, promote shared repositories, and advance real-time analytic tools to enable reproducible and clinically interpretable implementation [13].

Furthermore, current multimodal TMS–EEG research remains largely exploratory, with limited mechanistic linkage to molecular or hemodynamic processes. Future studies more systematically integrating TMS–EEG with complementary methodologies and within hypothesis-driven frameworks will help uncover convergent neurobiological mechanisms and enhance the translational relevance of identified markers. In parallel, integrating clinical, neurophysiological, and computational metrics within stratified or dimensional approaches will be key to delineating biologically homogeneous subgroups.

In addition, novel TMS–EEG metrics require advanced technical expertise, additional instrumentation, and yield more complex outcomes. For this reason, many recent studies continue to rely on traditional TEP measures (Table 1). In this respect, the integration of AI–based approaches may help overcome some of these obstacles by streamlining data processing and supporting the interpretation of high-dimensional datasets, particularly when combined with explainable AI frameworks that enhance model transparency and clinical interpretability.

Crucially, future studies will need to determine whether advanced TMS–EEG measures provide more consistent, reproducible, and clinically meaningful insights than traditional TEP metrics. To this end, multicenter collaborations and the open sharing of data and analytic pipelines, together with standardized inclusion criteria and validation protocols, will be key to ensuring replication across cohorts [70]. Promising directions that have not yet been applied in MDD include, but are not limited to, immediate TEPs [71,72], multimodal TMS-EEG-fMRI [73], and closed-loop stimulation protocols [74]. Ongoing methodological innovation in these areas has the potential to significantly advance our understanding of MDD and enhance the clinical utility of TMS-EEG.

## Figures and Tables

**Figure 1 biomedicines-13-02474-f001:**
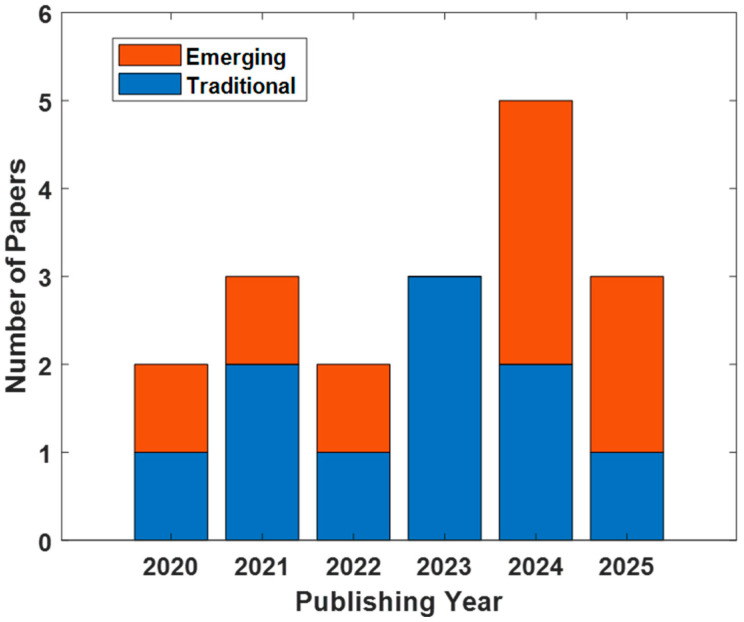
Publications about TMS-EEG and MDD in the period 2020–2025 (last search: 30 August 2025). In blue are the publications on traditional approaches (i.e., TMS-evoked potentials), in red are the publications on emerging approaches. The complete list of papers considered is provided in Table 1 and Table 2.

**Table 1 biomedicines-13-02474-t001:** Traditional TMS-EEG measures in MDD.

Reference Year	Authors	TEP Components	Sample	Stimulation Sites	Medications	Coil Type/Intensity (% rMT)
[17] 2020	Dhami et al.	N45, N100	45 MDD, 20 HC	Bilateral DLPFC, Motor Cortex, Inferior parietal lobule (IPL)	Stable treatment (medication or psychotherapy)	70 mm Figure-of-eight/120%
[18] 2021	Dhami et al.	N45	16 MDD, 16 HC	Bilateral DLPFC, Motor Cortex, IPL	Stable treatment (medication or psychotherapy)	70 mm Figure-of-eight/120%
[19] 2021	Voineskos et al.	N45, N100	30 Treatment-resistant depression (TRD)	Left DLPFC	yes	70 mm Figure-of-eight/120%
[20] 2022	Biermann et al.	N100	38 MDD	Bilateral DLPFC	yes	75 mm Figure-of-eight/120%
[16] 2023	Strafella et al.	N45, N100	185 MDD	Left DLPFC	yes	70 mm Figure-of-eight/120%
[21] 2023	Dhami et al.	P30, N45, P60, N100, P200	20 MDD-30 MDD	Bilateral DLPFC, bilateral IPL	Stable treatment (medication or psychotherapy	70 mm Figure-of-eight/120%
[22] 2023	Li et al.	P60	41 MDD, 42 HC	Left DLPFC	No info	70 mm Figure-of-eight/100%
[23] 2024	Li et al.	P180, P30	133 MDD, 76 HC	Left DLPFC	Stable medication	70 mm Figure-of-eight/100%
[24] 2024	Sheen et al.	N100	23 MDD	Right DLPFC	Stable medication	70 mm Figure-of-eight/100%
[25] 2025	Li et al.	P30, N45, P60, N100, P180, N280	59 MDD, 58 HC	Left DLFPC	Stable medication	70 mm Figure-of-eight/100%

**Table 2 biomedicines-13-02474-t002:** Emerging TMS-EEG metrics and approaches in MDD.

Reference Year	Authors	Metrics/Approaches	Sample	Stimulation Sites	Medication	Coil Type/Intensity (% rMT)
[26] 2020	Hadas et al.	Source-level analysis	31 TRD	Left DLPFC	yes	70 mm Figure-of-eight/100%
[27] 2021	Hill et al.	Oscillatory dynamics	38 MDD, 22 HC	Left DLPFC, Left primary motor cortex	yes	70 mm Figure-of-eight/120%
[28] 2022	Wada et al.	Multimodal/multiscale TMS-EEG approaches	60 TRD, 30 HC	Left DLPFC	yes	70 mm Figure-of-eight/120%
[29] 2024	Niu et al.	Trial-by-trial variability	34 MDD, 36 HC	Left DLPFC	yes	70 mm Figure-of-eight/110%
[30] 2024	Wada et al.	Multimodal/multiscale TMS-EEG	60 TRD, 30 HC	Left DLPFC	yes	70 mm Figure-of-eight/80, 120%
[31] 2024	Noda et al.	Machine learning approaches to TMS-EEG	60 MDD, 60 HC	Left DLPFC	yes	70 mm Figure-of-eight/-
[32] 2025	Zhang et al.	Microstate analysis	60 MDD, 60 HC	Left primary motor cortex	No (antidepressant)	70 mm Figure-of-eight/90%
[33] 2025	Chen et al.	Source-level analysis	166 MDD, 61 HC	Left DLPFC	yes	70 mm Figure-of-eight/100%

## Data Availability

No new data were created or analyzed in this study.

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
