# Peer review of "Unveiling Major Depressive Disorder Through TMS-EEG: From Traditional to Emerging Approaches"

_biomedicines, 2025, doi:10.3390/biomedicines13102474_

Round 1
Reviewer 1 Report
Comments and Suggestions for Authors
This review synthesized TMS-EEG studies from 2020–2025 to evaluate how traditional TMS-evoked potential (TEP) metrics and newer multidimensional approaches contribute to understanding MDD. It summarized consistent TEP findings implicating altered inhibitory/excitatory markers in the DLPFC and treatment-related modulation, then highlights emerging methods, including time–frequency and oscillatory analyses, microstate characterization, trial-by-trial variability, source-level reconstruction, multimodal/multiscale integration (including virtual histology), and machine learning, that exploited the full spatiotemporal and spectral richness of TMS-EEG to reveal network-level dysfunction, neuroplasticity signatures, and potential diagnostic and predictive biomarkers. The authors concluded that while TEPs remain valuable for methodological consistency, advanced analytic frameworks and AI-driven integration offer superior sensitivity for patient stratification and treatment monitoring, though technical complexity, heterogeneity across samples, and the need for larger, validated cohorts limit immediate clinical translation.
This review addresses highly timely and promising methodologies and is excellent; while I have no major objections to the manuscript overall, I recommend that the authors address or revise the following points.
Concerns and limitations:
- Clinical and biological heterogeneity of MDD: Study samples show substantial variability in diagnostic subtypes, illness stage, psychiatric and medical comorbidity, medication status, and demographic characteristics, which undermines biomarker specificity and compromises the validity of group-level inferences.
- Cross‑sectional designs and limited longitudinal evidence: The literature is dominated by cross‑sectional comparisons and short pre–post paradigms that preclude causal inference, do not establish temporal stability, and limit claims about prognostic validity of candidate markers.
- Lack of standardization in acquisition and preprocessing: Heterogeneity in TMS parameters (coil type, coil location, stimulation intensity, number/timing of pulses), EEG hardware and montages, artifact‑removal pipelines, and referencing schemes impedes reproducibility and prevents reliable aggregation or meta‑analysis across studies.
- Limited external validation and replication: Few results have been replicated in independent cohorts or tested in multicenter datasets, increasing the risk of cohort‑specific effects and overfitting of proposed biomarkers.
- Interpretability gaps for novel indices: The physiological bases of several emerging metrics (for example, eigenvalue‑based TTV measures, specific microstate transition profiles, and many ML‑derived feature sets) remain insufficiently characterized, hindering mechanistic interpretation and clinical translation.
- Confounding effects of medication and concurrent treatments: Incomplete control or reporting of psychotropic medication, convulsive therapies, and prior neuromodulation exposures introduces systematic confounds that can modulate TMS‑EEG responses and bias biomarker estimates.
- Insufficient multimodal integration and mechanistic linkage: Existing multimodal efforts are fragmented; systematic, hypothesis‑driven linkage of TMS‑EEG markers with cellular, molecular, or hemodynamic measures is scarce, limiting the capacity to infer underlying pathophysiological mechanisms.
- Practical and resource barriers to clinical implementation: The high technical complexity of TMS‑EEG, requirements for specialized hardware and expert personnel, lengthy preprocessing and analysis workflows, and the absence of robust normative databases restrict the feasibility of routine clinical deployment.
Reviewer 2 Report
Comments and Suggestions for Authors
The manuscript “Unveiling Major Depressive Disorder Through TMS-EEG: from traditional to emerging approaches” has been reviewed by me. The review paper provides a complete analysis of modern TMS-EEG applications for Major Depressive Disorder (MDD) by showing the evolution from basic TMS-evoked potential studies to advanced multi-dimensional methods. The research topic matches the current needs and the paper follows a proper organizational structure. The research needs substantial revisions to achieve better scientific quality and practical application of its findings. The review fails to assess methodological weaknesses in studied samples through evaluation of sample size variations and medication control and stimulation parameter differences. The addition of a table which presents essential methodological variations between studies will improve the review's understanding. The paper needs to include a thorough examination of TMS-EEG analysis reproducibility problems and the absence of established analysis protocols. The evaluation of emerging techniques remains valuable yet the study fails to include ablation experiments and method performance comparisons which prevents the determination of their individual effectiveness. The absence of public data access and code availability together with unclear data-sharing practices makes it impossible to verify the results presented in the paper. The authors should create a performance comparison table that shows traditional and new metrics including sensitivity and specificity and predictive accuracy values when available. The authors need to explain how explainable AI methods can enhance the clinical usability of machine learning models in their research. The authors need to specify required standardized inclusion/exclusion criteria and propose validation methods for different study cohorts in the section about limitations and future directions.
Round 2
Reviewer 2 Report
Comments and Suggestions for Authors
The authors have completely addressed all my comments, and I have no further concerns. Therefore, I recommend accepting the paper.